# The Impact of Underground Logistics System on Urban Sustainable Development: A System Dynamics Approach

**Jianjun Dong [1], Yuanxian Xu [2,*], Bon-gang Hwang [3], Rui Ren [2] and Zhilong Chen [2,*]**

[1] School of Civil Engineering, Nanjing Tech University, Nanjing 211816, China; dongjj@njtech.edu.cn
[2] School of Defense Engineering, Army Engineering University of PLA, Nanjing 210007, China; renrui0801@163.com
[3] Department of Building, National University of Singapore, Singapore 117566, Singapore; bdghbg@nus.edu.sg
* Correspondence: yuanx.xu@gmail.com (Y.X.); chen-zl@vip.163.com (Z.C.)

**Abstract:** The lack of practical application and accurate benefit analysis, which are the prerequisites for each other, make it difficult to implement and promote the underground logistics system (ULS), although in theory people always recognize its advantages in sustainable improvement of urban transportation and logistics. This paper attempts to use the system dynamics (SD) method, based on the real-world simulation, to analyze the quantitative relationship between the implementation strategy of ULS and the sustainability of urban transportation and logistics to solve this problem. Beijing city, China, was selected as the empirical background. Four ULS implementation strategies were proposed according to the city's potential investment in ULS and its demand for ULS network capacity. Meanwhile, four representative indicators were selected to evaluate the simulation results, including the average speed of the road networks in the peak hour, congestion loss, delivery travel time in the peak hour and the PM emissions of the truck. Good fitting index of historical data shows the validity of the model. Simulation results show that ULS, as a supplement to the urban integrated transport system, can significantly improve urban traffic and logistics. This study provides a perspective in the systematic and quantitative analysis of ULS to support the urban sustainable development.

**Keywords:** underground logistics system; system dynamics; urban sustainable development; benefit analysis

---

## 1. Introduction

Urban sustainable development and the sustainability of urban freight transport are interdependent with each other. However, the increasing negative externalities of the urban freight transport, such as exhaust pollution and congestion, seriously restrict the sustainable development of both [1]. Urban freight transport is a significant cause of traffic congestion, although the number of vans accounts for only 20–30 percent of total traffic [2]. Meanwhile the increasingly severe urban traffic congestion also seriously affects the efficiency of urban logistics. Freight transport activities are responsible for nearly 50 percent of the pollutants in urban air. Additionally, energy consumption and land utilization face enormous challenges due to the growth of urban freight demand [3].

Underground logistics systems (ULS) have been widely recognized as an effective means to solve these problems [4]. ULS transports goods via dedicated underground tunnels or pipes. Goods are automatically sorted and packed at the ULS underground station, and after that, they will be delivered to the customer through electric transport vehicles within the service scope. This new type

of infrastructure can directly reduce the amount of ground trucks to alleviate the negative impact of the freight. Meanwhile, it also can run at high speed 24 h a day to improve the logistics capability and efficiency of the city. Therefore, ULS is a more effective and sustainable approach than that which is adopted currently, such as traffic restrictions and adopting energy-saving truck, to alleviate freight traffic problems.

However, despite the great economic and social benefits, ULS still did not get a wide application because of the low social cognition degree [5]. Previous studies focused on ULS locomotive technology development and conceptual design [6,7]. A significant number of studies analyze the feasibility of the implementation of the ULS [8,9]. Nevertheless, few studies have systematically and quantitatively analyzed ULS comprehensive benefits and ULS how to affect the sustainable development of the city. Moreover, similar to the subway system, ULS can only achieve the ultimate benefits after the formation of the network. Hence, the impacts of the ULS should be evaluated over time. Undeniably, the development of ULS requires substantial investments and continuous policy support.

As a new type of transportation infrastructure, ULS has close interaction with urban development. On the one hand, network scale and density determine the transport capacity of ULS, and thus affect its social-economic benefits [10]. The implementation of ULS with high transport capacity can obviously greatly improve the efficiency of urban logistics. The transport capacity of ULS is inversely proportional to the number of ground transport vehicles, which directly affects congestion and traffic pollution. On the other hand, the network density of ULS will depend on policy support and investment strategies, and the local governments often play a dual role [11]. Their decision-making is based on the cost–benefit evaluation of ULS.

System dynamics (SD) has the advantage of combining qualitative and quantitative analysis to understand the internal mechanisms of a system. Meanwhile, SD can investigate the long-term variation trend of the system by changing the relevant variables of the decision-making. SD is particularly suitable for providing a system thinking for complex system with interactive factors, uncertainty, and time-related variables. The research object of this paper fits these characteristics.

With the aforementioned in mind, this study aims to clarify the relationship between the implementation of the ULS and the urban sustainable development by the system dynamics method, which is wildly used to quantitatively analyze the overall relationship among multiple systems and variables. Based on the simulation of real-world statistics in Beijing, China, the performance of ULS implementation strategy under different network densities is compared with the current transportation mode, so as to propose the effective ULS strategy conducive to the sustainable development of the city.

This study differs from previous research in two ways. First, ULS is seen as a potential complement to, rather than a replacement for, the current freight model. Second, the impacts of the ULS on the urban sustainable development are investigated comprehensively, both from the externalities of the urban freight transport, such as ecology and ground transportation, and the logistics operation efficiency.

The remainder is structured as follows: Section 2 presents a literature review focused on the ULS and the application of system dynamics in the urban sustainable development. Section 3 develops a system dynamics model to simulate the effectiveness of the ULS. Section 4 describes the simulation analysis. Discussion and conclusions are presented in Sections 5 and 6.

## 2. Literature Review

In this section, recent studies of ULS have been reviewed from three facets: research topics, cost–benefit and policy support. Meanwhile, the application of the system dynamics has been analyzed in the field of the urban sustainable development.

*2.1. Review of the ULS*

2.1.1. Research Topics

ULS related studies are mainly distributed in traffic engineering, industrial engineering, urban planning, and other disciplines. These studies demonstrate two patterns. The first stream focuses on the role of ULS in logistics and supply chain, and the second concerns its infrastructure attributes. Despite a substantial body of ULS–related research in multidisciplinary fields, what is lacking is a quantitative explanation of the impact of ULS on urban sustainable development. Existing studies lay particular stress on locomotive technology development and conceptual design of the system. However, they still play an important role in depicting the outline of ULS and constructing the theoretical model of this study.

Type of transport and locomotive—Various ULSs have been designed based on different types of underground transportation technology or locomotive, such as pneumatic capsule pipeline (PCP) [12], CargoCap [13], Dual Mode Trucks technology [14], Pipe§net [15], and AGV technology [16]. In these numerous ULSs, many innovative technologies have been widely applied, such as magnetic levitation technology, vacuum piping technology, linear introduction motors technology, and pneumatic transport system. Meanwhile, different types of the automated transport vehicles were invented with various size and functions to meet the transport needs. Therefore, the technical characteristics of the ULSs are different and applicable to different types of goods. However, the logistics process of various ULSs is consistent. The whole logistics process of ULS—such as warehousing, picking, and distribution—is operated underground, automated, and separate from passenger transport.

Conceptual design and network—In recent years, more attention has been paid to the conceptual design and network layout of ULS. ULS was implemented and designed as a single line in most studies. For example, an 80-km ULS was planned for the Ruhr region of Germany [17]. An ULS line in New York has proved technically and economically viable and will provide services to transport solid waste, parcels, and containers [18]. There are also routes in the Netherlands [19], Italy [20], Japan [21], and China [22]. However, the aforementioned potential applications were primarily concerned with reducing freight cost, the benefits of eliminating the ground truck problem have not been systematically calculated, which is one of the major concerns of the government. Mathematical models are also applied to ULS network design, such as the AHP-Fuzzy evaluation [23], and plant growth simulation algorithm [24]. In order to reduce the cost, the implementation of ULS based on urban subway system has also been widely discussed in recent years [10]. However, the models were still based on cost and freight efficiency, rather than the tradeoff between freight performance and freight transport externalities.

2.1.2. Cost–Benefit

There is no doubt that the benefits of ULS are huge, which is mentioned in almost every study. However, few articles systematically and specifically discussed its benefits. Qualitative, non-systematic and static analysis is the main state of ULS cost–benefit research. ULS can not only improve the capacity and efficiency of urban logistics, but also have great environmental and social benefits, which is a common concern in the research.

Cost—The construction of ULS is a complex underground engineering project, which requires huge construction investment. Its freight capacity and network scale are the first consideration [25]. Hence, it is necessary to establish a quantitative evaluation model to analyze the real impact of the scale of ULS network scale on urban development.

Environmental benefit—The environmental benefits of ULS are mainly reflected in the reduction of the number of ground trucks and thus the reduction of exhaust emissions. Although there is no relevant research on ULS, the literature on truck emissions can be used for reference. For example, Zahed used data from Texas to study the relationship between the reduction of vehicle mileage and the reduction of air pollutants [26]. The monetary value of reduction could reach 19,499 million dollars.

Social benefit—ULS has a wide range of social benefits, such as alleviating traffic congestion, improving road safety, reducing land use, and even improving regional vitality and value. However, the research seems to focus more on its advantages in alleviating traffic congestion, which is the biggest driving force for the implementation of ULS. The vans of Royal Mail were reduced to three-quarters after the adoption of ULS, thus improving the safety of the urban traffic and effectively alleviating congestion [27]. Zahed has investigated the monetary value of an ULS project in Texas, with dollars for the traffic congestion alleviating about 2423.6 million dollars and accident alleviating about 3194.75 million dollars [26]. However, existing studies generally lack the consideration of the urban traffic over time, such as the expansion of the urban road network and the increase of traffic volume.

### 2.1.3. Policy Support

Considering the huge social benefits and investment of ULS, it belongs to the infrastructure system with certain public properties [28]. Therefore, the government has the responsibility to promote the implementation and application of ULS. Moreover, the initiative of the government can play a leading role in attracting the participation of private capital [29]. Over the years, the lack of policy support has been the root cause of many ULS project failures, such as OLS-ASG [20] and so on.

In recent years, with the United States [30], France [31], Singapore [32], and China [33] have launched the feasibility study of ULS project, the situation gradually improved. The initial project license is clearly insufficient to support the actual implementation of the ULS project. This requires a range of policies, but research is sorely lacking, such as the degree of government participation, operation mode, price compensation mechanism, and so on. Of course, investors and policymakers are far from convinced that ULS have become as necessary to urban development as the subway system. Therefore, it is indispensable to comprehensively assess the benefits of ULS on the social, environmental, and urban logistics efficiency so as to facilitate the mental shift of the policymakers.

### *2.2. A Review of System Dynamics*

System dynamics (SD) is a commonly used and effective method to analyze urban development problems, first defined by Forrester in 1956 to analyze business management issues [34]. Several studies have used SD method to simulate the damage of urban congestion to urban development and analyze the effectiveness of existing policies [35,36]. A common conclusion is that, apart from directly reducing the amount of ground traffic, other policies have little effect on alleviating urban traffic congestion, such as reducing the use of the private cars and encouraging the choice of public transportation. Meanwhile, traffic pollution is also effectively reduced through the development of public transport system, the improvement of road network and the reduction of travel demand [37].

SD is also an effective means to quantitatively analyze the impact of logistics policies. Simulation results of logistics related policies show that increasing investment in transportation infrastructure and logistics supporting industries can effectively reduce the cost of urban logistics [38]. On the contrary, restrictive policies have seriously damaged the interests of the logistics industry [39]. Almost all current policies are unfriendly to truck carriers [40].

Given the above, this paper uses the SD model to analyze the comprehensive benefits and dynamic contributions of ULS, which has innovative theoretical and practical significance for clarifying the effectiveness of ULS system in urban development.

## 3. Model Development

The development of SD model mainly consists of three stages [41]. First, the variables of the system are identified according to the research purpose. Second, the causal loop diagram (CLD) is designed to analyze the causal relationships between multiple independent variables, which could be the goal or key driver of the system. Third, the stock-flow diagram is established to transform the qualitative relationship into the quantitative algebraic equations for policy analysis and simulation.

## 3.1. Empirical Background Analysis

The SD model needs real data for simulation to verify its effectiveness, and policy analysis is conducted based on the simulation results [42].

As the capital of China, Beijing is one of the fastest economic growing and most densely populated cities in the world. In the first three quarters of 2018, the delivery express business in Beijing has reached 1586.7 million pieces with an operating income of 24.1 billion RMB [43].

In this study, the Capital Core Functional Area of Beijing was selected as the research object. The Capital Core Functional Area refers specifically to the central area of the city, with an area of about 1,381 square kilometers, accounting for 8% of the total area of Beijing. The permanent resident population is about 12.4 million. According to the investigation of the China Academy of Transportation Sciences [44], the freight volume of the city center accounts for about 21% of the city's total, available on the website of Beijing Municipal Bureau of Statistics (http://www.bjstats.gov.cn/tjsj/) [43]. This is consistent with the published data of Beijing Transport Institute (http://www.bjtrc.org.cn/) [45].

## 3.2. Causal Loop Diagram (CLD)

The CLD in this study is divided into four parts, such as regulator, demand, ULS network, and performance, as shown in Figure 1. The variables of the feedback loops are linked by arrows indicating causality. Meanwhile, each arrow is marked with a polarity to indicate the positivity or negativity of the influence. The positive polarity ('+') means that the two variables will change in the same direction, and negative polarity ('−') means the opposite [46].

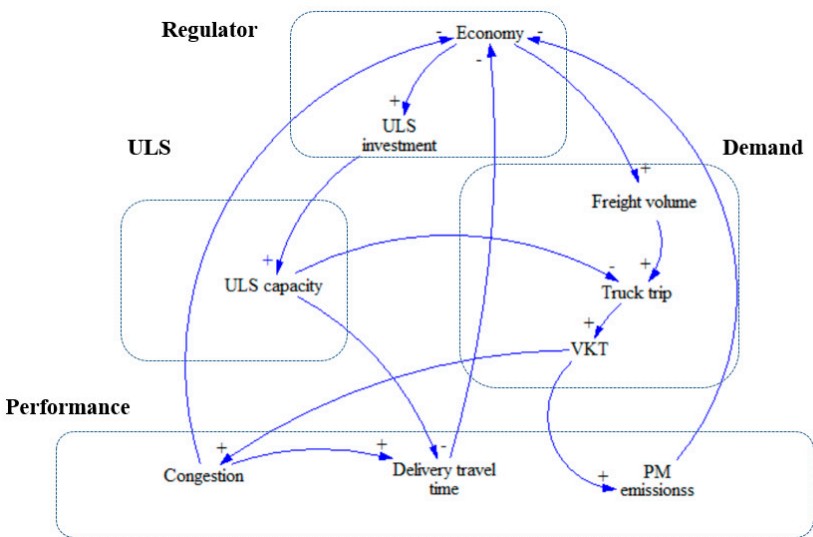

**Figure 1.** Causal loop diagram.

The 'Regulator' category contains two variables, economy and ULS investment, which are the key to logistics demand and ULS development. 'Demand' is driven by social development, which consists of freight volume, truck trip and vehicle kilometers travelled (VKT). 'Performance' is largely affected by 'Demand'. With the continuous growth of the urban freight volume, congestion, air pollution, increasing logistics delivery travel time and other problems are becoming serious, which have adverse effects on social development. 'ULS' is implemented to solve these problems, reducing the ground truck trip and the logistics delivery travel time with efficient operation.

Table 1 lists the seven feedback loops. Three balancing loops represent the negative impacts of serious congestion, increased PM emissions and the growing delivery travel time. Four reinforcing loops mean that the ULS can alleviate the current transport and logistics problems, which is consistent with the CLD.

**Table 1.** Feedback loops in the CLD.

| Category | Variables |
|---|---|
| Reinforcing Loop 1 | Economy→+ULS investment→+ULS capacity→−Truck trip→+VKT→+Congestion→+Delivery travel time→−Economy |
| Reinforcing Loop 2 | Economy→+ULS investment→+ULS capacity→−Truck trip→+VKT→+ PM emissions→−Economy |
| Reinforcing Loop 3 | Economy→+ULS investment→+ULS capacity→−Truck trip→+VKT→+Congestion→−Economy |
| Reinforcing Loop 4 | Economy→+ULS investment→+ULS capacity→−Delivery travel time→−Economy |
| Balancing Loop 1 | Economy→+Freight volume→+Truck trip→+VKT→+Congestion→−Economy |
| Balancing Loop 2 | Economy→+ Freight volume→+Truck trip→+VKT→+PM emissions→−Economy |
| Balancing Loop 3 | Economy→+ Freight volume→+Truck trip→+VKT→+Congestion→+ Delivery travel time→−Economy |

*3.3. Simulation Model Formation*

The system dynamics model—including stock, flow, shadow, and model variables—is developed to examine the dynamic influences of ULS on the urban sustainable development. Stock variables symbolized by rectangles represent the state of the system and are affected by the flow variables over time. Flow variables depicted as valves in the model represent the rates of the growth. Only the name of model variables is displayed in the system. The system is always divided into multiple small parts called views. Shadow variables, described as '< >', are employed to represent the model variables in the other views. Vensim PLE 7.2a software (Ventana Systems, Inc., Harvard, USA), commonly used for the SD simulation, is adopted for the validation and simulation. According to the results of CLD division, the model could be divided into four subsystems: regulator subsystem, demand subsystem, ULS subsystem, and performance subsystem. Their details are explained in the followed sections. Table 2 lists a number of variables and their abbreviations that appear in the model for easy reading and lookup.

**Table 2.** Main variables and abbreviations of the model.

| Notation | Variable | Reference | Notation | Variable | Reference |
|---|---|---|---|---|---|
| UFV | ULS freight volume | [10] | CL | Congestion loss | [47] |
| UC | ULS capability | [10] | ATTT | Average truck travel time | [48] |
| UITF | ULS investment proportion table function | [10] | DTPH | Delivery travel time in the peak hour | [48] |
| UNPP | ULS network project implementation progress | [10] | GDP | GDP | [49] |
| FV | Freight volume of the city | [10] | GDPGR | GDP growth rate | [49] |
| FVCC | Freight volume of the city center | [10] | GDPT | GDP increase | [49] |
| ULD | ULS logistics demand | [10] | TTV | Truck trip volume | [50] |
| UND | ULS network density | [10] | ATTD | Average truck trip distance | [50] |
| AUTT | Average ULS travel time | [10] | TPHF | Truck trip peak hour factor | [50] |
| UFP | ULS freight proportion | [10] | TVCF | Truck vehicle conversion factor | [50] |
| UI | ULS investment | [10] | TI | Traffic intensity | [50] |

**Table 2.** *Cont.*

| Notation | Variable | Reference | Notation | Variable | Reference |
|---|---|---|---|---|---|
| UULC | ULS network unit logistics capacity | [10] | RC | Road capacity | [50] |
| UN | ULS network | [10] | GT | Ground transportation | [50] |
| UR | ULS logistics supply and demand ratio | [10] | AS | Average speed | [50] |
| TT | Truck turnover | [44] | TLI | Transportation and logistics investment | [50] |
| FVRF | Freight volume reduction factor | [44] | VKT | Vehicle kilometers travelled | [50] |
| DT | Delay time | [47] | PME | PM emissions | [51] |
| VPT | Value of person time | [47] | EF | Emission factor | [51] |
| | | | TL | Truck load | [52] |

### 3.3.1. Regulator Subsystem

Figure 2 shows the regulator subsystem, which describes the impact of urban economic development on the investment of ULS.

Equation (1) is conductive to calculate the GDP stock, in which the flow variable *GDPI* represents GDP increase and *INTEG* means the function of integral. $GDP_{t0}$ and $GDP_{tn}$ indicate the GDP in the start time and termination time, respectively.

*GDPI* is calculated by Equation (2), in which *GDPGR* represents the GDP growth rate. The related data can be gathered from the Beijing Statistical Year Book (2007–2016) [43].

$$GDP_{tn} = INTEG(GDPI, GDP_{t0}) \tag{1}$$

$$GDPI = GDP \times GDPGR \tag{2}$$

ULS investment (*UI*) depends on the total transportation and logistics investment of the city (*TLI*). The development strategy of ULS will also affect *UI*, which can be expressed as the ULS investment proportion table function (*UITF*) in Equation (3).

$$UI = TLI \times UITF \tag{3}$$

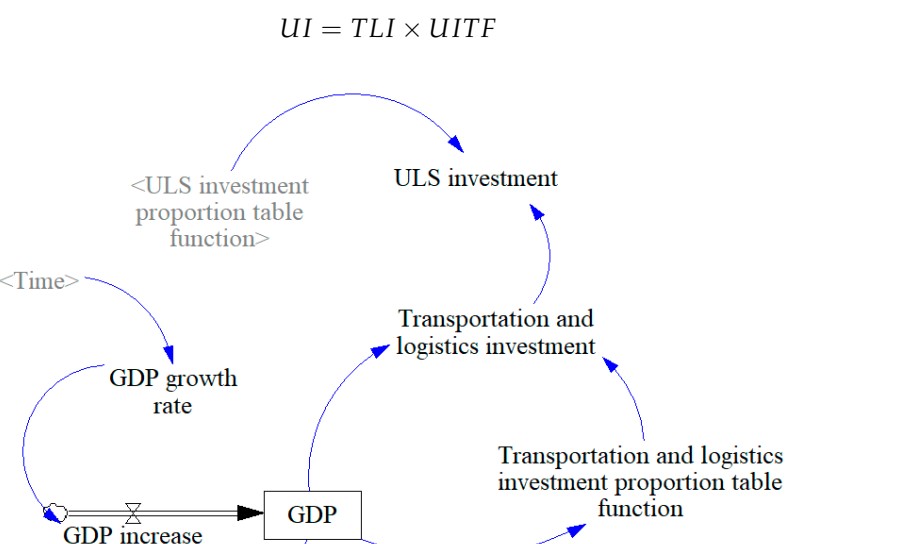

**Figure 2.** Regulator subsystem.

### 3.3.2. Demand Subsystem

Figure 3 shows the demand subsystem, which illustrates the problems and demand of urban traffic and logistics. In general, the contradiction between the increasing demand for transportation and freight in cities and the deteriorating traffic condition has caused great economic losses. In this subsystem, the average speed of the road networks in the peak hour is selected to represent the traffic condition of the city. The ULS could significantly reduce truck trips and increase average speed of the road networks.

The traffic condition is evaluated by the ground transportation (*GT*), which is defined as the vehicle turnover in the peak hour. *GT* mainly takes into account the impacts of private car, taxi, bus and truck. Taking truck as an example, the truck turnover (*TT*) depends on the truck trip volume (*TTV*), the truck trip peak hour factor (*TPHF*), the truck vehicle conversion factor (*TVCF*), and the average truck trip distance (*ATTD*), as shown in Equation (4). *TVCF* is introduced to convert trucks into standard vehicles for calculation, as the impact of the truck on congestion can be equivalent to the impact of several private cars [50,53]. Days and orders of magnitude are also considered for the model.

$$TT = TTV \times ATTD \times TVCF \times TPHF/3.65 \tag{4}$$

*TTV* is influenced by the freight volume of the city center (*FVCC*), the ULS freight volume (*UFV*) and the truck load (*TL*), as shown in Equation (5). The implementation of the ULS, expressed as the *UFV*, will decrease the *TTV* and further affect the ground traffic.

$$TTV = (FVCC - UFV)/TL \tag{5}$$

*FVCC* is the function of the freight volume of the city (*FV*) and the freight volume reduction factor (*FVRF*), as shown in Equation (6). The *FV* is obtained by using SPSS software for data fitting, depicted in Equation (7).

$$FVCC = FV \times FVRF \tag{6}$$

$$FV = 162.876 \times GDP^{0.507289} \tag{7}$$

The turnover of private car, taxi, and bus can be calculated by using the similar method, and *GT* is the sum of these four different transport modes.

Traffic intensity (*TI*) is defined as the ratio of the *GT* to the road capacity (*RC*) to investigate the condition of the urban transport demand and supply, as shown in Equation (8). *RC* is defined to be the vehicles turnover of the urban roads. Average speed (*AS*), defined as the average speed of the road networks in the peak hour, is one of the main indicators of urban traffic characteristics, which can be calculated by Equation (9) [50,54].

$$TI = GT/RC \tag{8}$$

$$AS = 125.19 \times EXP(-0.7866 \times TI) \tag{9}$$

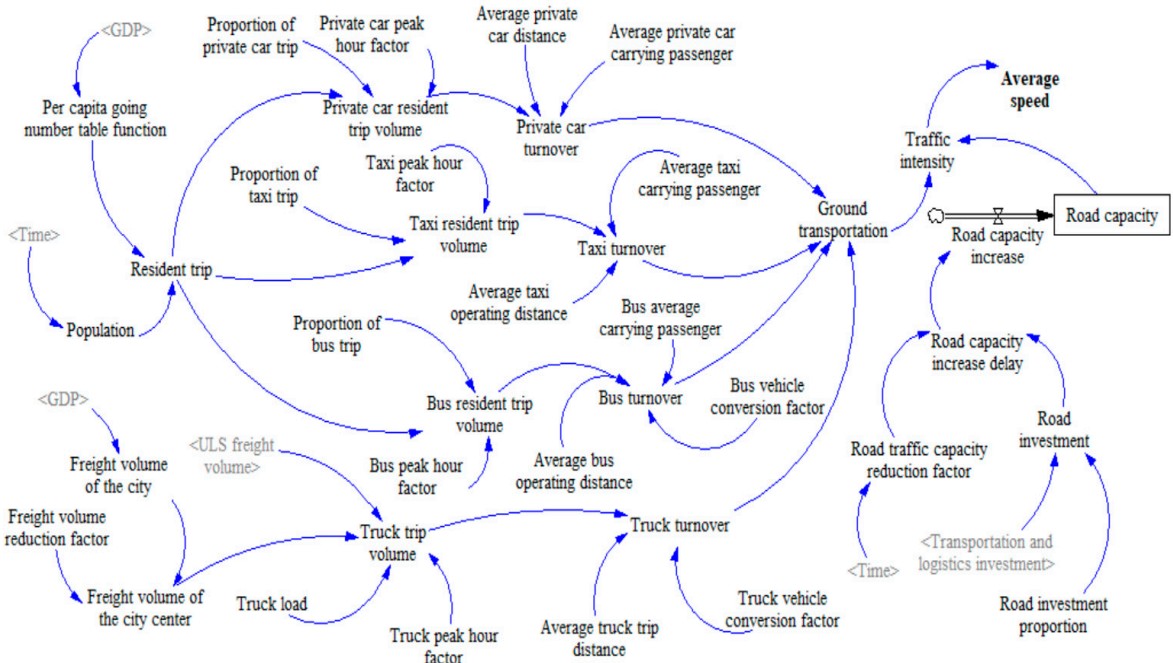

**Figure 3.** Demand subsystem.

### 3.3.3. ULS Subsystem

As mentioned before, the utility of the ULS significantly depends on the network scale and the actual amount of goods that pass through it. Figure 4 shows the ULS subsystem built around two important characteristic variables of ULS, network density (*UND*) and system capability (*UC*). The *UND*, defined as the ratio of the area served by ULS to the area of the entire planning area, will affect the ULS investment strategy, which is expressed as the *UITF*. The *UITF* is shown in Equation (10), where *WITH LOOKUP* means a table function that can establish a functional relation between variables and dependent variables through a data table.

$$UITF = WITH\ LOOKUP(UND, (UNDt0, UNDtn)) \tag{10}$$

*UC* is proportional to *UND*, and assuming that the ULS only transport goods within a reasonable service area. Accordingly, *UC* is the function of ULS network (*UN*) and ULS network unit logistics capacity (*UULC*), as shown in Equation (11). *INTEGER* in the equation means an integer for the variable.

$$UC = INTEGER(UN) \times UULC \tag{11}$$

The logistics demand of the ULS served area is expressed as the ULS logistics demand (*ULD*), which can be considered as a function of the *UND*, the *FVCC* and the ULS network project implementation progress (*UNPP*), as shown in Equation (12). *UNPP* reflects the construction progress of the ULS.

$$ULD = UNPP \times UND \times FVCC \tag{12}$$

In addition, since the regional logistics demand may exceed the capacity of ULS, especially in the early stage of network construction, the *UFV* is introduced to represent the actual freight volume transferred underground, which is the smaller value of *ULD* and *UC*, as shown in Equation (13). Meanwhile, the ratio of the *UFV* to *ULD* is represented as the ULS logistics supply and demand ratio (*UR*), as shown in Equation (14).

$$UFV = MIN(ULD, UC) \tag{13}$$

$$UR = UFV / ULD \tag{14}$$

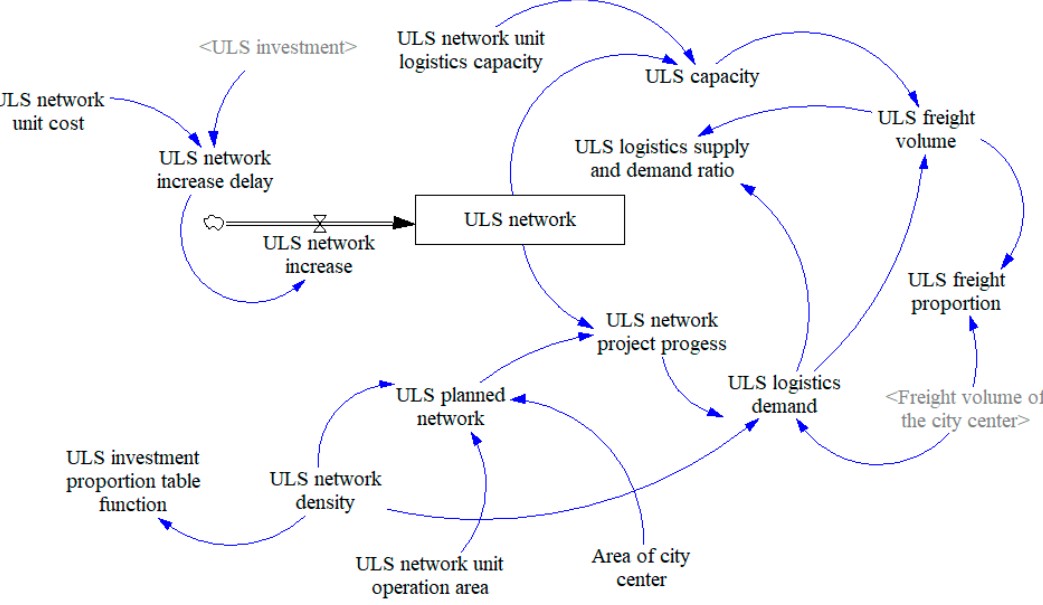

**Figure 4.** ULS subsystem.

### 3.3.4. Performance Subsystem

ULS performance subsystem is constructed by selecting three main benefits of ULS, decreasing logistics delivery travel time, alleviating congestion and reducing PM emission, as shown in Figures 5–7, respectively.

Congestion occurs when the average traffic speed on urban roads is slower than the ideal speed, resulting in massive economic losses for passengers using all modes of transportation due to time delays. Reducing the congestion loss (*CL*) is used to measure the performance of ULS in alleviating traffic congestion. The consequences of traffic time delay are various, and the calculation of its value loss is complex and lacking unified standards. In this study, the value of delay time for passenger is estimated, and the equation of the *CL* refers to the report from the Texas A&M transportation institute [47]. The *CL* is related to the delay time (*DT*) and the value of person time (*VPT*), as shown in Equation (15). The data related to *VPT* refers to the report from the CHINA ECONOMIC WEEKLY [55]. Days are also considered for the model.

$$CL = DT \times VPT \times 365 \tag{15}$$

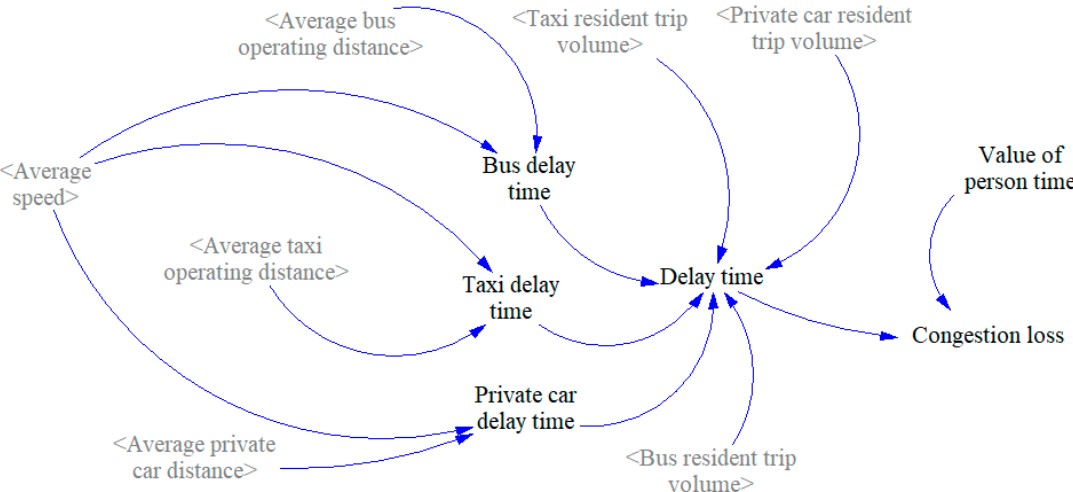

**Figure 5.** Flow diagram of congestion loss.

PM from vehicle exhaust is considered the unhealthiest pollutant in metropolitan areas, such as PM$_{2.5}$ and PM$_{10}$. Trucks are the main potential source of PM pollution, and the PM emissions can be calculated by the vehicle kilometers travelled (*VKT*) [50]. In addition, the PM emissions are also influenced by the emission factor (*EF*), which has been changed from Euro III to V. The variable of the PM emissions (*PME*) is depicted in Equation (16). Round-trip distance of the truck and days are also considered in the model.

$$PME = EF \times VKT \times 730 \tag{16}$$

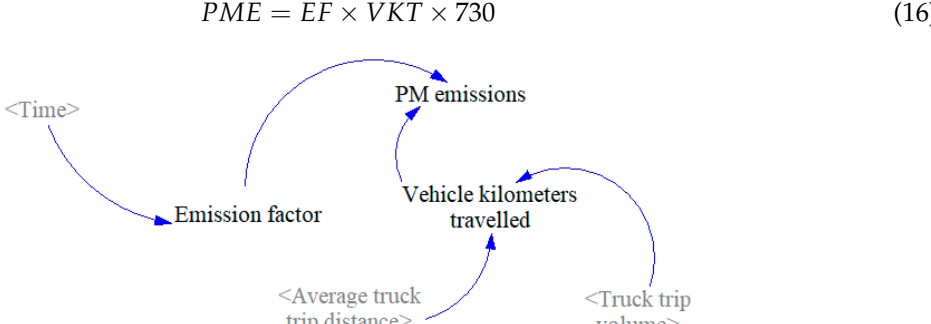

**Figure 6.** Flow diagram of PM emissions.

Delivery travel time, which is commonly used to evaluate the efficiency of the logistics operation [48,53], is significantly related to the congestion, especially in the peak hour [56]. Therefore, the delivery travel time in the peak hour (*DTPH*) is a more direct indicator when measuring the utility of ULS for congestion. It is determined by the goods travel time both on truck and in the ULS. The average truck travel time (*ATTT*) is related to the average driving speed in the road network, while the average ULS travel time (*AUTT*) is constant and unrelated to the ground traffic. The average driving speed of automated transport vehicles for the ULS ranges from 40–60 km/h [17], and its median value is 50 km/h for calculation in this paper. The *DTPH* is calculated in Equation (17) with the weighted average value of *ATTT* and *AUTT*. The ULS freight proportion (*UFP*), defined to be the ratio of *UFV* to *FVCC*, is introduced to represent the weight ratio, as shown in Equation (18).

$$DTPH = (1 - UFP) \times ATTT + UFP \times AUTT \tag{17}$$

$$UFP = UFV / \text{FVCC} \tag{18}$$

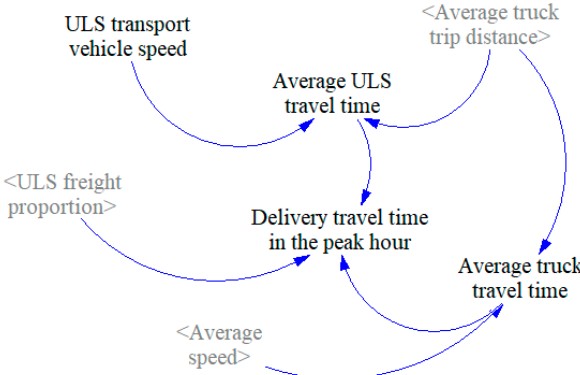

**Figure 7.** Flow diagram of delivery travel time in the peak hour.

## 4. Data and Model Testing

Model testing is a key step before the quantitative simulation and analysis. There are a lot of predictions and assumptions in the model, and the ULS system is not really implemented, so we used the real data from 2007 to 2016 to calibrate the parameter setting and verify the model. The basic data were obtained from Beijing Statistical Year Book (2007–2016) [43] and Beijing Urban Traffic Annual

Report (2007–2016) [45]. Table 3 shows the values of the main variables. The model test is divided into three steps: output fitting of historical data, the behavioral validity test, and then the extreme conditions test. The main variables of the model, such as the *AS* and the *FV*, have been selected for the model test.

**Table 3.** Value of main variables in the simulation.

| Variable | Value/Unit |
| --- | --- |
| Freight volume reduction factor (FVRF) | 0.211 |
| Truck peak hour factor (TPHF) | 0.2 |
| Value of person time (VPT) | 7.5 dollars/h |
| Truck load (TL) | 1 ton/vehicle |
| Average truck trip distance (ATTD) | 30 km |
| Truck vehicle conversion factor (TVCF) | 3 |
| ULS transport vehicle speed | 50 km/h |
| ULS network unit logistics capacity (UULC) | 0.33 million tons/year |
| ULS network unit operation area | 4 square meters/unit |
| ULS network unit cost | 59.7 million dollars/unit |
| Proportion of private car trip | 0.33 |
| Proportion of taxi trip | 0.06 |
| Proportion of bus trip | 0.27 |
| Private car peak hour factor | 0.2 |
| Taxi peak hour factor | 0.1 |
| Bus peak hour factor | 0.16 |

## 4.1. Output Fitting of Historical Data

First, the output fitting of historical data was analyzed to verify the validity of the model. The comparison between the *FV* and the actual data in Figure 8 shows that the simulation results well reflect the historical value from 2007 to 2014. However, the plots do not fit fairly well between 2015 and 2016. The actual *FV* decreased obviously, whereas the simulation results still maintain an upward trend. Due to the enforcement of environmental protection policies, a large number of enterprises located in the outskirts of the city have been forced to relocate, resulting in a large decline in *FV* [57]. However, these policies have little impact on the *FVCC*, and the simulation results are still effective.

In addition, Table 4 shows the percentage error between the actual and simulated values. The results show that the percentage error from 2007 to 2014 less than 5% meets the commonly testing standard, and the model has a high confidence [42].

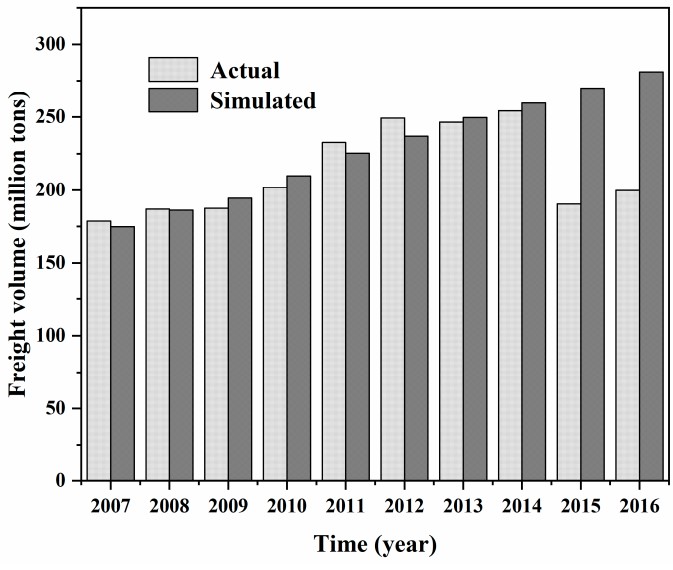

**Figure 8.** Historical and simulated freight volume of the city.

**Table 4.** Error of the historical and simulated freight volume of the city.

| Year | 2007 | 2008 | 2009 | 2010 | 2011 | 2012 | 2013 | 2014 | 2015 | 2016 |
|---|---|---|---|---|---|---|---|---|---|---|
| Historical values (million tons) | 17872 | 18689 | 18753 | 20184 | 23276 | 24925 | 24651 | 25416 | 19044 | 19972 |
| Simulated values (million tons) | 17482 | 18608 | 19440 | 20981 | 22537 | 23694 | 24958 | 25954 | 26975 | 28097 |
| Error (%) | 2.2% | 0.4% | 3.7% | 3.9% | 3.2% | 4.9% | 1.2% | 2.1% | 41.6% | 40.6% |

*4.2. Behavioral Validity of the Model*

Then, the behavioral validity was verified through the qualitative and quantitative comparison between the model-generated behavior and the empirical behavior, followed by Sterman [58] and Gómez [59]. The comparison of the actual and simulated *AS* is illustrated in Figure 9. A visual inspection of the plot suggests that the simulated models satisfactorily capture the trend of the *AS*. In general, *AS* in this region is very low, never exceeding 28.5 km/h. Before 2013, the *AS* shows an upward trend partly due to the expansion of urban road network. After that, the curve of *AS* becomes smooth and gradually drops. The rapid growth of *FVCC* had brought great pressure on ground transportation and slowed down the average driving speed in urban road network.

Besides, a quantitative assessment of the fit between the simulated and the actual *AS* was conducted by calculating mean average percentage error (MAPE), mean square error (MSE), as well as Theil inequality statistics, as shown in Table 5. Theil inequality statistics provide an MSE decomposition of bias ($U^M$), unequal variation ($U^S$) and unequal co-variations ($U^C$). The sum of $U^M$, $U^S$ and $U^C$ always equals 1. The results indicate low $U^M$ and $U^S$ and a variance concentrated in $U^C$. This implies that the simulated variables provide a good fit for the actual data, and the errors are unsystematic. Moreover, the high $U^C$ reveals that the model and the data have the same phasing, but different amplitude fluctuations. That is because the study focused on long-term behavior and ignored short-term fluctuations.

Therefore, the behavioral validity testing further verified the validity of the model, which can effectively replicate the historical behaviors and capture historical trends.

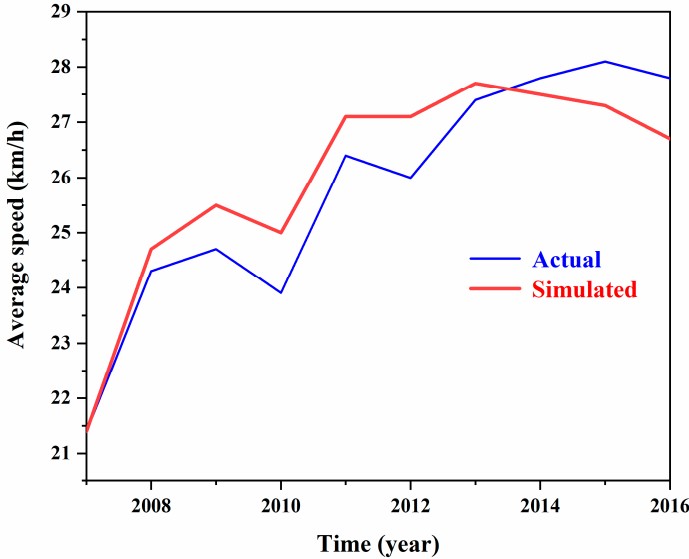

**Figure 9.** Historical and simulated average speed.

**Table 5.** Error analysis of the average speed.

| | MAPE (%) | MSE (units$^2$) | $U^M$ | $U^S$ | $U^C$ |
|---|---|---|---|---|---|
| **Average speed** | 2.5 | 0.57 | 0.08 | 0.09 | 0.83 |

### 4.3. Extreme Conditions Test

Finally, the extreme conditions test was conducted following the approach in Sterman to check the robustness of the model by assigning extreme values to model parameters [60]. Set the parameter of *FV* to 5 times and 0.2 times of the original parameter value respectively to simulate *AS* in extreme cases of high freight volume (HFV) and low freight volume (LFV), as shown in Figure 10. Meanwhile, the original case is called the base case. In the case of LFV, the FV is significantly reduced, thus the traffic condition is improved and the *AS* is increased. On the contrary, the high freight volume demand increases the road networks burden and reduces the speed of traffic. Meanwhile, the simulation results exhibit a similar trend compared with the base case. Therefore, a good description of extreme cases shows the rationality of the model performance.

In conclusion, the validity and robustness of the model were verified. The simulation results are reliable and consistent with the historical data. The model can reflect the real world and serve as an effective instrument for the next policy simulation.

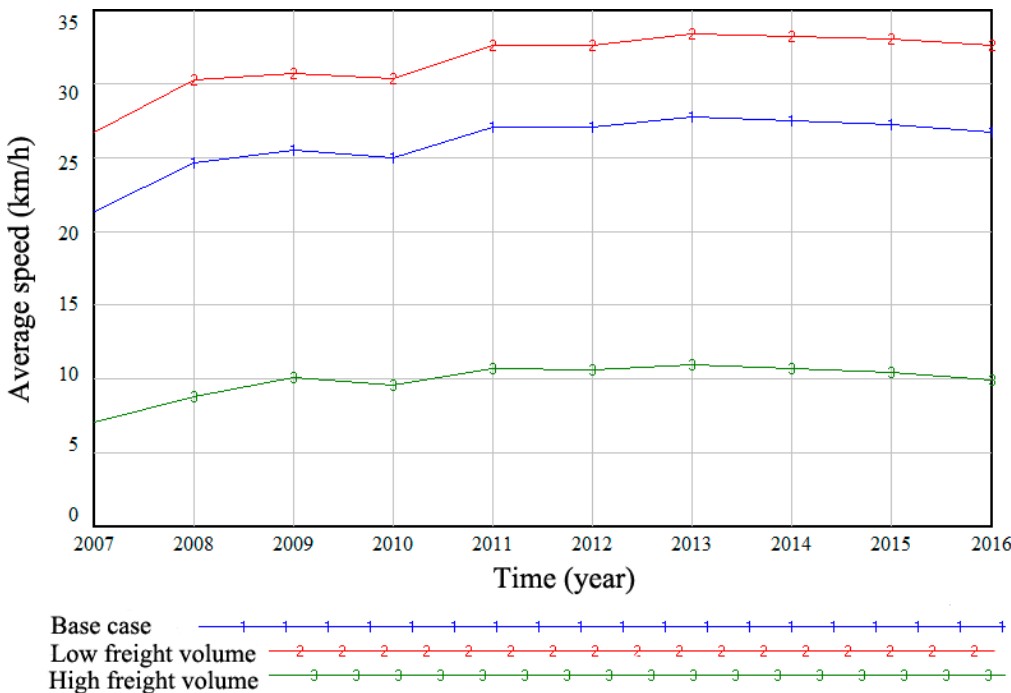

**Figure 10.** Extreme conditions test of the average speed.

## 5. System Simulation and Results

### 5.1. Implementation Strategies of ULS

The implementation strategy of ULS mainly considers the total amount of goods allocated underground to determine the transportation capacity of the proposed ULS, which is finally reflected in the network density of ULS. This study designed a set of ULS simulation strategies including four types of different network densities—called base case (BC), low density (LD), medium density (MD), and high density (HD)—as shown in Table 6. Strategy BC indicates that the network density is zero, i.e., ULS is not adopted. Meanwhile, the performance of the strategy was measured by the variables of *AS*, *CL*, *DTPH*, and *PME*. Furthermore, assuming that the construction time of the whole network is 10 years, from 2017 to 2026, the capacity of the system will reach its maxim value in 2027. Meanwhile, the construction time of each unit needs two years and the system will be operational from 2019. In this way, the *UC* increases year by year with the expansion of the network. Accordingly, the traffic and logistics situation in the central urban area of Beijing from 2017 to 2035 was predicted and simulated.

**Table 6.** Parameters of Beijing ULS.

| Variable | BC | LD | MD | HD |
|---|---|---|---|---|
| ULS network density | 0 | 0.4 | 0.6 | 0.8 |

### 5.2. Simulation Results and Analysis

Figure 11 shows the simulation results of *UC* and *ULD* in the scenario of HD. It suggests that the *UC* increases with the expansion of the ULS network and can meet most of the logistics demand in the ULS covered area. Until 2024, *UC* is greater than *ULD*, which means that all goods can be transported through the ULS. However, the transport demand exceeds the limit of *UC* from 2025. ULS can still transport 78.2 million tons of goods per year and the *UR* reaches as high as 80.6% in 2035. This result means that the ULS is an effective supplement for the current transportation mode. It can significantly alleviate the great burden brought by the rapid growth of the urban freight volume. The similar trend can be found in the scenarios of LD and MD.

The investment of the ULS is determined by the *UND* and increased with the expansion of the ULS network. The total investment reaches to about 5.7, 8.6, and 11.5 billion dollars, respectively, for the LD, MD, and HD.

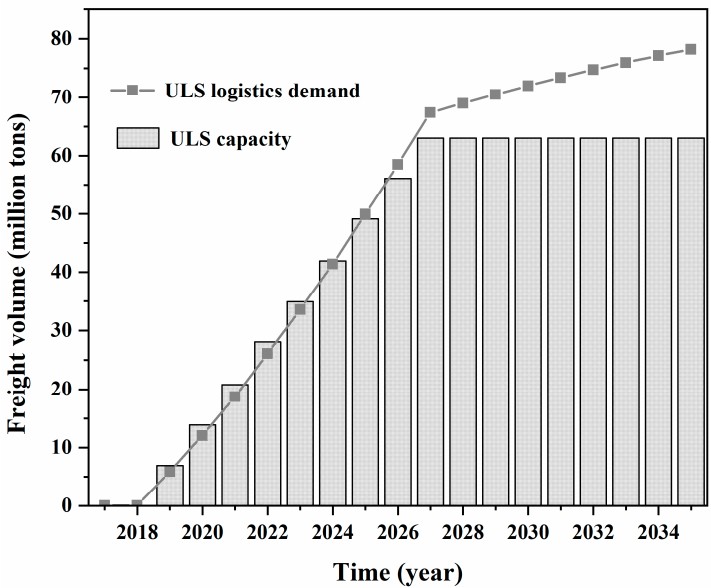

**Figure 11.** Comparison of the ULS capacity and the ULS logistics demand in the scenario of HD.

Figure 12 shows the simulated results of the *TTV*. The *TTV* will reach 98.2 million trips by 2035, under BC. However, an obvious decrease occurs with the application of the ULS, and the *TTV* declines to 66.8, 50.7, and 35.2 million, under the strategies of LD, MD, and HD, respectively. Among the strategies, the HD is best at the reduction of the *TTV*. *TTV* is decreased by the growth of the *UC* even the ULS is in full load from 2025. However, two years later, with the finish of the expansion of the ULS network in 2027, the *TTV* starts to grow fast and deteriorates the ground traffic condition.

Figure 13 presents a series of simulation results for verifying the performance of ULS implementation strategy. The overall results show that the traffic and logistics situation of the city center will continue to deteriorate rapidly in the future if it develops according to the current situation. The implementation of ULS can effectively alleviate these problems, but the degree of relief is closely related to the network density of ULS.

Figure 13a shows the simulation results of *AS* in the empirical region under the designed strategy set. When the ULS is not implemented, that is, under the BC strategy, the *AS* will drop to 13.5 km/h in the peak hour by 2035. However, the situation would be obviously improved with the implementation

of the ULS. In 2035, *AS* under LD, MD, and HD strategies are 15.3, 16.4, and 17.3 km/h, respectively, which increased by 13.3, 21.5, and 28.1% compared with BC. Under the condition of HD, ULS effectively alleviated the deterioration of *AS* until 2027. Since the simulated logistics demand has exceeded the design capacity of ULS since then, truck traffic on the ground is once again active, which leads to the deterioration of *AS*.

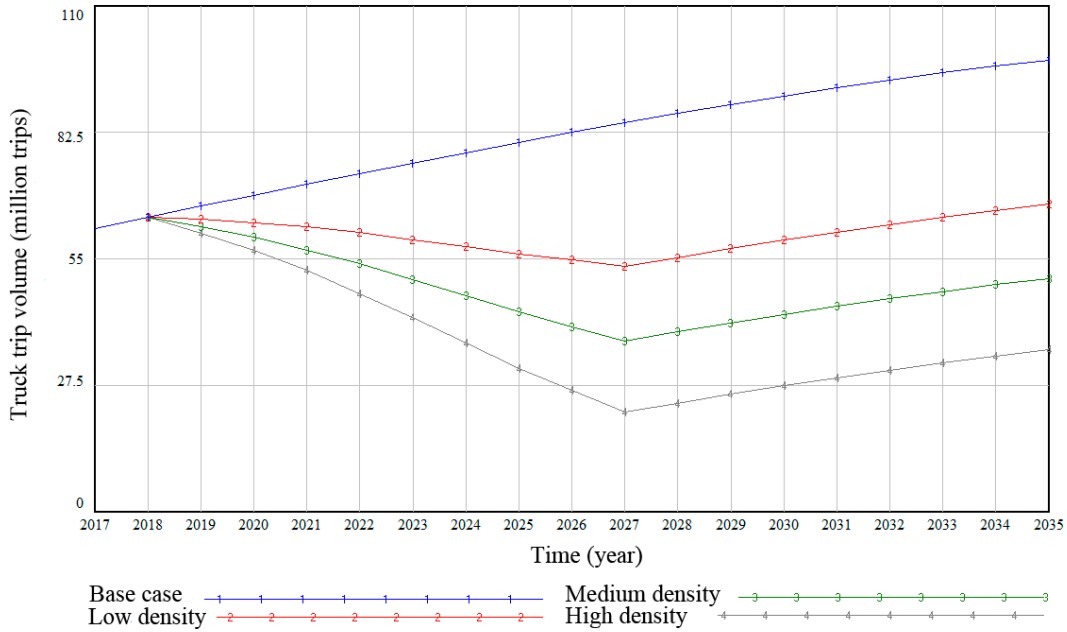

**Figure 12.** Simulation results of the truck trip volume.

Figure 13b shows the simulation results of the *CL*. By 2035, the *CL* under the BC strategy will reach 14.3 billion dollars. Nevertheless, it would be largely reduced to 12.2, 11.2, and 10.3 billion dollars, respectively, in the case of the LD, MD, and HD. Compared with BC, the congestion loss reduction is about 14.7, 21.7, and 28.0%, respectively. The performance of the HD is the best, while the LD is the worst.

Congestion would lead to the increase of travel time and the uncertainty of distribution, which will have a negative impact on urban logistics efficiency. Figure 13c simulates the *DTPH*. The rapid growth of the travel time can be seen from the BC curve. The delivery time of the BC would be as high as 2.2 h in 2035. Contrarily, the downward trend appears when the ULS has been implemented. The delivery travel time would decline to 1.5, 1.2, and 1.0 h in the case of LD, MD and HD, respectively. The HD has the best performance. In 2027, there is also a trend of performance failure caused by the full load of ULS.

The simulation results of the *PME* are depicted in Figure 13d. In the case of BC, *PME* will increase sharply and reach 111.9 t in 2035. Similar to the previous index, after the implementation of ULS, the emissions of the LD, MD, and HD strategies are 76.3, 57.8, and 40.1 t, respectively, which are 31.8, 48.3, and 64.2% lower than those of BC.

On the whole, simulation results show that the implementation of ULS has significant benefits on urban transportation and logistics. However, its contribution to alleviating traffic congestion is far less than that to improving logistics efficiency and reducing PM emissions. Of course, this is related to the impact coefficient of trucks set in the model on road congestion. Trucks emit far more exhaust than cars, but they make up a small proportion of road vehicles. If the number of empty trucks is added, the simulation results will be more satisfactory. In fact, data show that the proportion of traffic, pollution and accidents caused by empty trucks transporting on roads is between 15% and 30% [61]. In addition, the network capacity of ULS will far exceed the value set in the model by improving locomotive traction and reasonable operation organization and even expanding the diameter of the

underground tunnel. Therefore, from a long-term perspective, the implementation of ULS has high benefits for urban sustainable development.

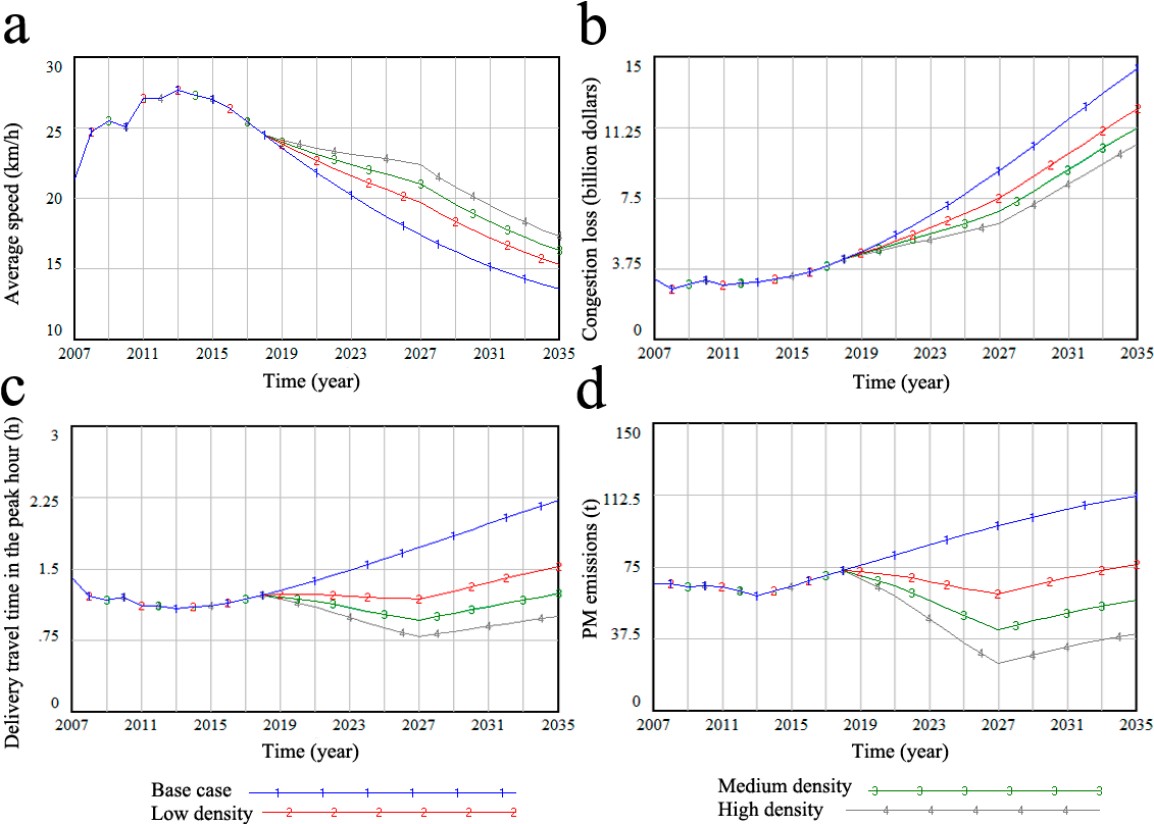

**Figure 13.** Simulation results of the model. (**a**) Average speed; (**b**) Congestion loss; (**c**) Delivery travel time in the peak hour; (**d**) PM emissions.

## 6. Conclusions

This study investigates the implementation strategies of ULS and its impact on the urban sustainable development. In the process of ULS research and application, unclear benefits have always been an important obstacle to its large-scale implementation. Good social benefits provide good incentives for the development of ULS. However, the lack of practical verification and quantitative simulation analysis leads to overstating the social benefits of ULS and neglecting its other functions as an urban logistics infrastructure. In addition, ULS does not simply reduce the number of ground trucks to gain benefits, it is closely related to network density and urban logistics demand. Therefore, it is necessary to analyze the complex relationship between ULS and its social and economic benefits based on demand forecasting.

This study has developed a SD model to analyze the impacts of the ULS on the urban sustainable development based on the empirical background of the central region of Beijing city. ULS implementation strategies under three network densities are proposed, and ULS performance is evaluated by comparing traditional logistics strategies. With the adoption of the ULS, truck traffic volume can be significantly decreased, thereby alleviating congestion and reducing truck-related PM emissions. In addition, the logistics delivery travel time is greatly reduced due to the high efficiency of the ULS and the improvement of urban transportation. The simulation results show that the high-density development strategy will achieve the best performance. The four performance indicators selected were significantly improved. This paper has comprehensively analyzed the positive effects brought about by the implementation of the ULS, which is conductive for policy makers to have a deep cognition of the ULS. In particular, ULS can significantly promote the urban logistics efficiency, which

not only provides an original perspective for the planning of the ULS network, but also offers a great incentive for the policy makers to pay more attention on the improvement of the logistics industry by ULS. Furthermore, of specific interest in this study is the incorporation of both market demand and investment decisions, along with the effects of interactive decisions on the performance outcomes. Although the investment of the ULS is relatively high, considering its great social and environmental benefits, ULS can be considered as a high-return scheme from the perspective of the government. Systematic and quantitative analysis of the advantages of the ULS contributes to the positioning and development of ULS, as well as for the formulation of relevant policies.

Several limitations inevitably appear in this study, and also represent valuable directions for future research. First, SD simulation does not provide detailed analysis of the network formation process. The long construction period is a characteristic worthy of consideration for underground engineering. Second, there are many indirect benefits or variables worth discussing between direct benefits and urban sustainability, such as land appreciation and road safety. Third, this study only simulates the benefits of ULS symbolically. Combining the construction cost and price compensation policy, further exploring the improvement performance of ULS on urban logistics is the next research direction.

**Author Contributions:** R.R. contributed to data collection; J.D. and Y.X. proposed the research framework, analyzed the data and wrote the article; B.-g.H. and Z.C. contributed to revising article.

**Funding:** This study was supported by the National Natural Science Foundation of China (grant no. 71631007).

**Conflicts of Interest:** The authors declare no conflict of interest.

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
