# Peer review of "The Impact of Underground Logistics System on Urban Sustainable Development: A System Dynamics Approach"

_sustainability, doi:10.3390/su11051223_

Round 1

Reviewer 1 Report

This paper applies system dynamic method to underground logistic system. This approach is new in transportation field and their analysis have enough contributions. I have only minor comments.

1. Since the one of main contributions in this study is application of the SD methodology, it should be discussed/introduced in the first (introduction) section.

2. Even though the methodology of this study is interesting, the conclusion seems not new. For example, “With the adoption of the ULS, truck traffic volume can be significantly decreased, thereby alleviating congestion and reducing truck-related PM emissions” and “the logistics delivery travel time is greatly reduced due to the high efficiency of the ULS and the improvement of urban transportation” are just as it should be. I wonder how this study provides new insights to policy makers.

Author Response

    Thank you for the reviewer’s comments, and those comments are all valuable and very helpful for revising and improving our paper, as well as the important guiding significance to our researches. We accept and appreciate the comments of the reviewer. 

Reviewer 2 Report

It's really an excellent article. However, the cost of constructing the system has not been explained in any way; even the technological aspects deserve further investigation, especially as regards the connection with surface transport (last mile).

 There are some typographical errors:

 Line 16  China was           China, was

Line 86                Conclusions                       conclusions

Line 113              New York to has              New York too has  ??

Line 170              use                                      used???

Line 230              Table 1. Feedback loops in the CLD               I think there are problems in + and –

Line 401               Error of the historical and simulated freight volume of the city             add units (million tons)

Line 449               called BC                             specify acronym

Line 453               Furthermore, assuming that the construction time of the whole network is ten years, from

2017 to 2026, and each unit needs two years            something is missing??

Author Response

(The authors gave the same response as above.)
